# Lived experience of young people with epilepsy in Bahir Dar city government hospitals, Ethiopia, a qualitative interview study

**Kokeb Ayele[1], Habtamu Wondiye[2], Eyob Ketema Bogale**[ID][2]*

**1** Health Promotion and Behavioral Sciences Department, Wollo University, Dessie, Ethiopia, **2** Health Promotion and Behavioral Sciences Department, College of Medicine and Health Sciences, Bahir Dar University, Bahir Dar, Ethiopia

* ketema.eyob@gmail.com

**Data Availability Statement:** All relevant data are within the paper and its Supporting Information files

## Abstract

### Background

Epilepsy is the most common neurological disorder, which is characterized by persistent derangement of the nervous system due to an abrupt excessive discharge of the group of neurons from the cerebrum. For developing young people with epilepsy coping with the challenges of living with epilepsy and adjusting normative tasks associated with adolescence is stressful in all domains of the lives of young people with epilepsy. But in Ethiopia, published literature regarding the lived experience of young people with epilepsy is lacking.

### Objectives

This study aims to explore lived experience of young people with epilepsy.

### Methods

A qualitative interview study was conducted in Bahir Dar city government specialized and referral Hospitals, from February to April, 2021. Using the rule of saturation, a total of 11 study participants (age 12–24) were selected using hetrogenious types of purposive sampling technique. The data were collected through semi structured interviews technique with the aid of audio recorder. Semi-structured interview guide and observation checklist for care context in health facilities were used. The collected data was transcribed word by word and translated conceptually. The Data were analyzed using thematic approach. The credibility, dependability, Conformability and transferability of the study were assured using different techniques such as peer debriefing, member checking; audit Trail, thick description and purposeful sampling. Atlas- ti version7 software was used to facilitate data analysis.

### Results

The participants' narratives revealed two main themes: 'experiences due epilepsy' and 'coping strategies to wards epilepsy'. Experience due to epilepsy' was described by young people with epilepsy in terms of psychosocial, physical, economical and health care related

**Funding:** The author(s) received no specific funding for this work.

**Competing interests:** The authors have declared that no competing interests exist.

**Abbreviations:** AED, Anti-Epileptic Drug; AWE, Adolescent with Epilepsy; PWE, People with Epilepsy; BDU, Bahir Dar University; EFY, Ethiopian Fiscal Year; FHCSH, Felege Hiwot Comprehensive Specialized Hospital; MPH, Master of Public Health; HP, Health promotion; IBE, International Bureau for Epilepsy; ILAE, International League against Epilepsy; ERB, Ethical Review Board; OPD, Out Patient Department; QOL, Quality Of Life; TGSH, Tibebe Gion Specialized Hospital; WCL3RH, West command level three referral hospital; WHA, World Health Assembly; YLDs, Years Lived with Disability; YPE, Young People with Epilepsy; WHO, World Health Organization.

experiences. They described coping strategies towards epilepsy in terms of finding support from family and society as well as religious institutions and other traditions as coping strategy.

## Conclusion

Even though young people with epilepsy had suffered a lot of hurtful experiences, they reported coping strategies towards epilepsy that include support from various sources. These types of findings have implications for social work interventions for young people living with epilepsy.

## Introduction

Epilepsy is a neurological disorder that is characterized by persistent derangement of the nervous system due to an abrupt excessive disorganized discharge of the group of neurons from the cerebrum The excessive discharges cause a disruption of sensation, convulsive movement, or psychic function without or with loss of consciousness [1].

Globally, Epilepsy is the most common neurological disorder, affecting about 70 million people, the majority of whom (80%) live in resource-poor countries, where epilepsy remains a major public health problem because of its health implications as well as its social, cultural, psychological, and economic significance [2]. The World Health Organization (WHO) has recognized epilepsy as a public health concern that leads to an increased risk of premature death, increased healthcare needs and expenditure, and loss of work productivity [3,4]. Epilepsy ranks 20th as a cause of years lived with disability (YLDs), globally [5].

Defeating epilepsy is a global public health commitment and a new challenge. In 2015 the World Health Assembly (WHA) discussed the issue of epilepsy and adopted a resolution on the global burden of epilepsy and the need for coordinated action at the country level to address its health, social, and public knowledge implications [6].

In sub-Saharan Africa, the disease affects approximately 10 million people annually. Even if it affects all age groups its lifetime prevalence has bimodal peaks in young people and the elderly [7]. People with epilepsy in sub-Saharan Africa may experience severe isolation and discrimination in many areas of life, including the health care sector because epilepsy is often perceived as a curse, a mental illness, or a contagious disease [8].

In Ethiopia, the incidence and prevalence of epilepsy were reported to be 64/per 100,000 population and 520/per 100,000 population. The prevalence of psychiatric disorders such as anxiety and depression among PWE was found to be 33.5% and 32.8% respectively [9].

Adolescence is a period of transition from childhood to adulthood characterized by changes in physical maturity, emotional development, and other life-changing measures [10]. When a chronic illness, such as epilepsy, overlaps with the healthcare needs of a developing adolescent, an issue about the management of their condition becomes difficult. This challenging condition includes lifestyle adjustments, medication compliance, and other social situations that lead to anxiety-provoking for young people with epilepsy (YPE) and could interfere with normal developmental milestones, such as achieving independence. The cumulative effect of complex management of epilepsy as well as the rapid change associated with adolescent youth development results in compromised quality of life (QOL) for the YPE. In addition, the transitional years between childhood and adulthood are a particularly vulnerable period for YPE creating specialized healthcare needs [11].

Living with epilepsy experience extends far beyond medical seizure management. People with epilepsy (PWE) face a range of challenges in their life. This includes restricted independence, adverse effects of antiepileptic drugs (AEDs), stigma, and greater difficulty attaining educational and employment-related goals. Living with epilepsy also leads to psychological distress and poor quality of life. Poor health outcome for YPE leads to a life-long impact. This includes Increases in unemployment, binge drinking, unplanned pregnancy, and a decrease in high school graduation rates that have been documented in young adults with a current or past history of epilepsy [12].

PWE struggle to live and come to terms with their condition within the biomedical model's narrow construction of epilepsy, with its central focus on AEDs treatment and seizure control. Although, how the medical model diverts blame for seizure occurrence away from PWE, affords them a certain level of protection from their condition. It is full of dissatisfaction for PWE. Due to this, the medical construction of epilepsy as a seizure disorder causes tension for PWE. This leads to a poor health care system with unmet needs for emotional, social, and psychological health need [13]. Above all, exploring lived experience of young people with epilepsy has great value to achieve the resolution of the global burden of epilepsy.

People with epilepsy are also at increased risk of psychiatric comorbidity compared to the general population [14]. A previous report stated that young people with epilepsy suffered poorer psychosocial outcomes compared to their peers. Young people with epilepsy had emotional suffering such as stress, angry, loneliness, embarrassment, annoyed, frustrated, sadness [15].

In Ethiopia, a community study revealed that the majority of people with epilepsy were found to seek help from both religious and biomedical healing centres, with a 12 month treatment gap for biomedical care of 56.7% and a lifetime treatment gap of 26.9% [16].

As far as literature searching on Google Scholar, PubMed, and journal estimators about the issue under study showed, there are predominance quantitative studies in the general population [17–19]. However, there is no published work in Ethiopia regarding the lived experience of YPE. This shortage of information regarding the issue under study epilepsy in Ethiopia may hinder addressing psychosocial, physical and economic, and healthcare-related consequences caused by epilepsy.

Qualitative research may be best suited to explore it. The purpose of this qualitative interview study was to explore the lived experience of in YPE Bahir-Dar city government referral Hospitals for a deep understanding of their condition and the challenges they face in their day-to-day life and the impact of disease on their, psychosocial, physical, and economic health. It aims to give a voice for AWE to express their individual experience living with the disease epilepsy, to gain an understanding of the essence of the experience of living with epilepsy, and to provide a description of the lived experience.

## Methods and materials

### Study setting, design and period

**Approach and setting.** A qualitative interview study was conducted from February 28 to April 08, 2021. A qualitative Interview is a research approach used in Qualitative studies where more personal interaction is required and detailed data is gathered from the participant [20].

This study was conducted in government specialized and referral hospitals in Bahir Dar city, Ethiopia. Bahir-Dar is located approximately 578 km north-northwest of Addis Ababa. The city has three government-specialized and referral hospitals. These hospitals were Felege-Hiwot comprehensive specialized hospital (FHCSH), Tibebe Gihon specialized hospital (TGSH), and West Command level 3 referral hospital (WCL3RH). Felege Hiwot and Tibebe

Gion specialized hospitals are known to provide services on four major wards namely internal medicine, surgery, pediatrics, and gynecology and obstetrics. In addition to these services, it provides different services for different chronic diseases including Epilepsy. Felege Hiwot hospital and Tibebe Gion give epilepsy care at NCDs (Non-communicable diseases) clinic whereas West command level 3 referral hospital gives epilepsy care at a psychiatric clinic.

## Population and sample

We recruited a total of 11 participants from Bahir Dar city government specialized referral hospitals and data saturation was maintained. The participants were eligible for the study if they had been receiving care in the study clinic for at least 6 months duration since diagnosis. Seriously ill patients who are unable to communicate at the time of the data collection period were excluded from the study.

A heterogeneous type of purposive sampling strategy was used to recruit the participants. Purposeful sampling is a technique widely used in qualitative research for the selection of information-rich cases, individuals that are experienced with a phenomenon of interest. In addition, it considers the ability to communicate experiences and opinions in an articulate, expressive, and reflective manner [21].

A heterogeneous type of purposive sampling technique was used to recruit the participants. The heterogeneous characteristics of participants were maintained by considering age, marital status, educational status, health facilities, and residence, including the most vulnerable young people in the context of Ethiopian priorities and culture. This was done to be able to see the different forms of the lived experience of young epileptic patients from different perspectives.

**Data collection.**  For this study, the Interviewer was the key instrument for data collection and the interpreter of the findings [21]. Literature was reviewed to develop an interview guide [22]. The data were collected using pretested interview guide. The interview guide contains 12 socio demographic question and 17 epileptic patients experience related questions. Interviews were conducted in the Amharic language. The investigator developed an early familiarity with the culture of the selected health facilities. We made a list of the characteristics (heterogeneous) that our participants should have. Then we Identified and sampled every person who meets the sample criteria. The doctors and nurses working in the follow up clinics and admission ward at Felege-Hiwot comprehensive specialized hospital (FHCSH), Tibebe Gihon specialized hospital (TGSH), and West Command level 3 referral hospital helped in identifying patients that fulfill the inclusion criteria of the study. Then the participants were communicated with before or after they get treatment on follow-up or admitted room after they recovered from their condition.

When the participant agreed to participate in the study, a meeting was scheduled at a time that is convenient for the participant. The location ensured the participant's privacy and was mutually agreed upon, which was their home except two interviews were conducted in a separate room in the hospital.

The interview consisted of semi-structured questions, face-to-face in a quiet location. The interviewer requested permission to start the interview and audio recording to ensure verbatim transcription. The interview was conducted until conceptual saturation reached (to the point no further new information was obtained anymore). In addition, an observation checklist was used to collect care context data in health facilities

The probing technique was applied by using how and why, to get adequate data on the point of interest. Participants were coded as P1, P2, and P3. . .. respectively according to their order of interviewing. The range of interviewing time had taken from 30 to 60 minutes. The interviewer had taken notes during the interview, regarding body language, nonverbal cues and labeled the audio with the pseudonym.

The process of data collection continued until it reached the point of saturation. In turn, Data saturation was attained when all questions were answered, responses were completed and no further new information was obtained from various respondents. The data saturation was achieved in nineth interview and the additional two interviews were conducted for confirmation of data saturation.The interviewer thanked them after the completion of the interview.

### Ethics approval and consent to participate

Ethical approval for this study was obtained from the Institutional Review Board of Bahir Dar University (Approval number: CMHS/IRB/ 26/008/2021).The Institutional Review Board of College of Medicine and Health Sciences, Bahir Dar University decided and approved that verbal informed consent obtained from each study participant could be enough to be ethically assured of the research process. This was because unless the name and the participants' medical registration number (MRN) were used during data collection, there is no ethical issue that will be raised.

For young participants whose age was younger than 18 verbal informed consent was obtained from both participants and their legally authorized representatives before the study. Legally authorized representatives were given detailed information about the purpose of the study, data collection procedures, and possible risks and benefits of participating in the study through the consent process. Verbal informed consent was obtained from all legally authorized representatives whose children participated in the study. In this case, the families were presented as legally authorized representatives. A child was included in the study only if the families agreed with the child. Despite the families, consent, a child's decision not to participate in the study was respected.

Verbal informed consent was obtained from each study participant before the commencement of data collection. Participants also gave their verbal informed consent to share their data purely for scientific purposes without disclosing their true name.

**Data analysis.** After data collection, the principal investigator transcribed the audio-recorded data in the participant's local language in written form and then back-translated to English by a third party. Data obtained from observation were written after completion of each fieldwork every day. The thematic analysis approach was used to analyse the data. The six thematic data analysis steps were followed to analyze the data such as Step 1: Become familiar with the data, Step 2: Generate initial codes, Step 3: Search for themes, Step 4: Review themes, Step 5: Define themes and Step 6: Write-up [23].

The principal investigators were read and reread the transcriptions several times and hear the audio-taped interview repeatedly to provide a sense of integrity and understand the meaning of the experiences from the participant's viewpoint. Each meaning unit was labeled with a code representing its content by open coding and then similar code organized into categories. Atlas. ti software version 7 was used to facilitate data analysis. Two independent coders were participated During coding. Categories were peer-reviewed and checked by the co-lead author and final subthemes and themes were created. Lastly, the report was written as subthemes and themes based on objectives for presenting the discoveries of the study. Quotes were used to highlight each category and show association with each theme.

**Rigor and trustworthiness of the study.** This study was trustworthy based on Lincoln and Guba's criteria of credibility, dependability, conformability and transferability [20].

*Credibility*. Credibility means ensuring the results are believable, consistent with reality and that the interpretations are true [20]. Credibility was assured by keeping the consistency of procedures and the neutrality of the investigator about findings or decisions. Data collection sessions involved only those who are genuinely willing to take part and prepared to offer data

freely (Ensure honesty in participants). Data collection tools were pretested before the actual study in a similar setting and population to ensure that the interview length and clarity of guide were appropriate for the actual participants. In addition to this, the interview guide was checked for consistency and correctness by an expert in a psychiatric clinic. Probing and multiple data sources (combination of in-depth interviews and observation) were employed to collect the data. We also invited 6 participants to review the ideas for transcription verification, which they think the investigator is going to present a true picture from their perspective. The questionnaires, audio data, transcriptions, and findings were given to the peer (who had qualitative research experience) to cross-check the processes, evaluate the findings and get feedback (peer debriefing). Moreover, at the onset of the study, bracketing the preconceptions of the investigator was employed to minimize the investigator's bias and the risk of reactivity by the participants.

*Transferability*. Transferability means showing that the findings have applicability in other contexts [20]. To allow judgments about transferability by the reader appropriate probes had been used to obtain detailed information on responses. During each interview, the data collector observed participants' body movements; facial expressions, eye gaze, and tone of speech, and everything was recorded (persistent observations). Transferability was achieved by describing the study setting, sample, and data collection procedure clearly and using qualitative expert for peer debriefing.

*Dependability*. Dependability is the assessment of the data collection and data analysis process. Dependability was attained through accurate documentation by minimizing spelling errors through frequent check, including all documents in the final report such as including the notes were written during the interview and ensure that the details of the procedures were described to allow the readers to see the basis upon which conclusions was made [20] each of which was considered at all times.

*Conformability*. Conformability is a measure of how well the study's discoveries are supported by the data collected and reflects the objectivity of the data. This means that the researcher's bias should not alter the result [20]. Confirmability of the study was ensured by the recording of every activity of the participants during the time of the interview. In addition, the audio-taped interviews were not destroyed which can enable others to track the process. Moreover, data analysis, interpretations and conclusions of findings were shared with experienced qualitative researchers for peer debriefing before synthesizing the final outputs. It was also achieved by using quotes which means linking the words of the participants with the discoveries [20].

## Results

### Socio-demographic characteristics of respondents

A total of 11 in-depth interviews were conducted (Table 1).

### Themes and subthemes

The findings that emerged from the analysis of the in-depth interview are presented and arranged as themes, sub-themes and categories. Data analysis has revealed five themes that express lived experience of young people with epilepsy. The themes are: (1) Psycho-social experience with four sub themes such as feelings about living with epilepsy, social isolation, parent-imposed restrictions and academic difficulties/challenges. (2), physical experience with one sub theme which is recurrent physical trauma due to epilepsy. (3), economic experience with two sub themes such as direct cost of illness or out-of-pocket costs and indirect cost of illness i.e., productivity loss and unemployment/ job discrimination. (4), Health care related experience with two sub themes such as contexts related to Epileptic care and patient-provider

**Table 1. Socio-demographic characteristics of study participants among three government referral hospitals in Bahir Dar city, Ethiopia, 2021.**

| Characteristics of Respondents | Classifications | Number or frequency |
|---|---|---|
| Sex | Male | 7 |
| | Female | 4 |
| Age | 12 -13years | 1 |
| | 14 years | 2 |
| | 15–19 years | 3 |
| | 20–24 years | 5 |
| Residence | Rural | 8 |
| | Urban | 3 |
| Marital status | Under age/single | 8 |
| | Married | 1 |
| | Divorced | 1 |
| Educational status | Not attended | 2 |
| | Elementary | 5 |
| | Secondary and preparatory | 2 |
| | Diploma and above | 2 |
| Occupational status | Daily labour | 2 |
| | Student | 4 |
| | Civil servant | 1 |
| | No work | 4 |
| | Housewife | 1 |
| Religion | Orthodox | 9 |
| | Muslim | 1 |
| | Protestant | 1 |
| Number/category of drug they took | One drug | 5 |
| | More than one drugs | 6 |
| Controlled with anti-epileptic drug | Yes | 3 |
| | No | 8 |

relationship. (5) Coping strategies towards epilepsy which have two subthemes such as seeking support and religious institutions and other tradition as coping strategy (Fig 1).

## Theme 1: Psycho-social experience

More than half of the study participants faced at least one negative psychosocial experience relating to their living with epilepsy; the participants discussed these hurtful experiences in three subthemes as feelings about living with epilepsy, social isolation, and academic difficulties. these categories are interrelated to each other i.e., if young people with epilepsy suffered intense negative feelings such as social embarrassment, they may isolate themselves from society and they might have academic difficulties due to these hurtful negative feelings such as feeling fatigued and feeling sleepy. The reverse is also true in which having poor academic status, might result in intense negative feelings such as stress, loneliness, and feeling different which might lead to social isolation.

## Subtheme 1: Feelings about living with epilepsy

**Negative feelings.** Almost all of the respondents had some feelings about their epilepsy. Most of the participants often reported negative feelings such as stress, anger, loneliness, embarrassment, annoyance, frustration, sadness, feeling different from their friends, and tired

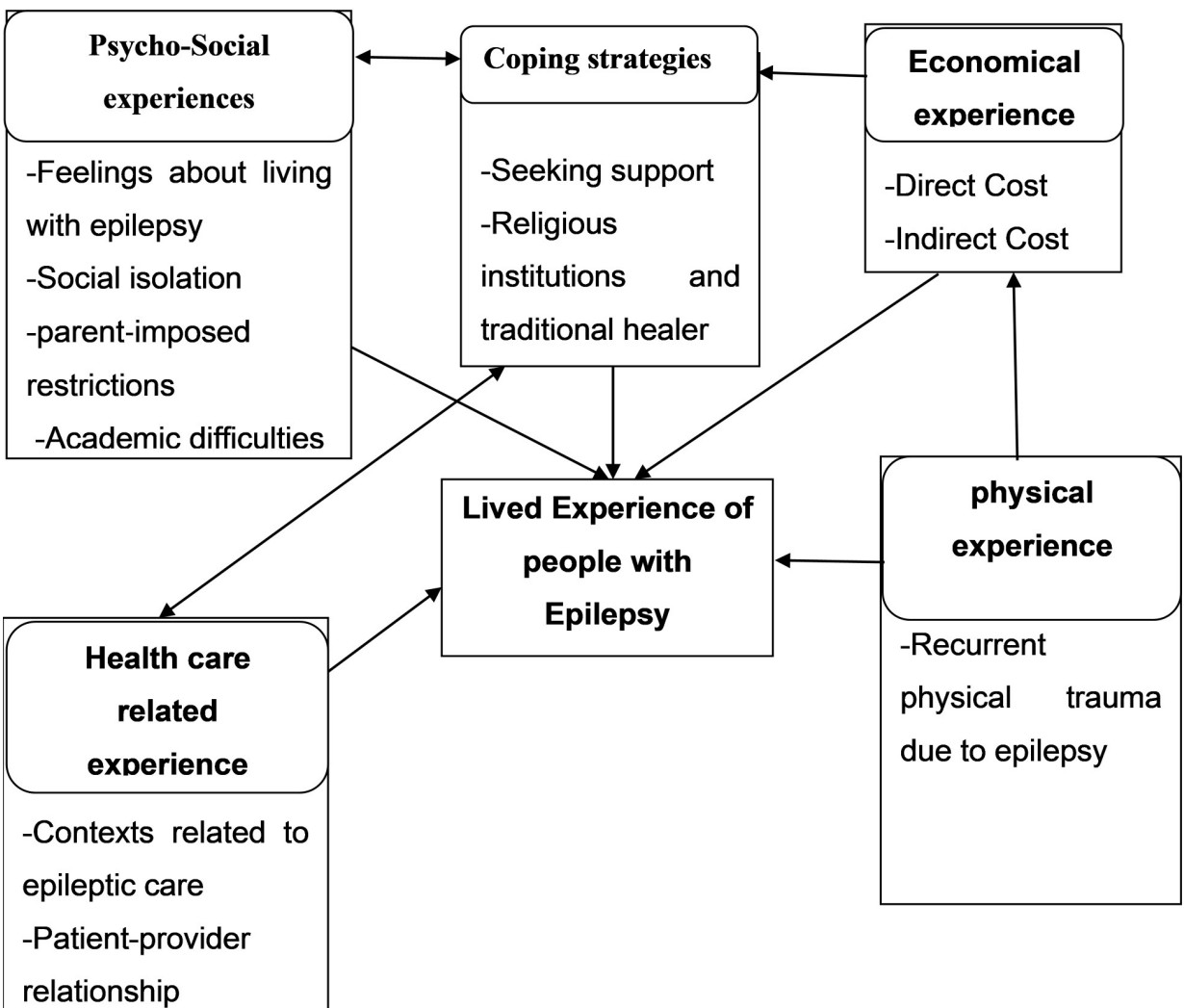

**Fig 1. Themes and subthemes for lived experiences of Ethiopian young people with epilepsy, in Bahir-Dar city, Ethiopia, 2021.**

and sleepy. They also expressed fears about injury and death. As the participants' narratives report revealed these feelings were either due to the unpredictable nature of the seizure itself, or situations connected to it such as restricted activities, taking anti-epileptic drugs, and reduced levels of participation.

The unpredictable nature of the seizure: the occurrence of a seizure is unpredictable which means it happens suddenly at any time in any place including in a public area result embarrassment or it may happen in a risky environment that endangers the life of the patient

The following quotes showed that the negative feelings were due to its sudden happening.

*"It [epilepsy] is a very terrifying disease and I worried and feel distrust every time because it is sudden happening and it made me feel sad and tired. . . "14 –years-old male respondent*

On the other hand, another participant's narrative revealed that the negative feelings were mostly due to restricted activities caused by being an epileptic patient. This youth's depression and anxiety lead him to suicidal ideation as his narrative reveals

*"I feel worried because it [epilepsy] prevents me from the job. I always think that is why this happened to me. Why I am usually absent from my job, why I was inferior to my relatives and friends. Why almighty God only made me ill. . . I always think badly about the future sometimes I feel killing myself [suicidal Ideation] . . ."* 23-year-old male respondent

In addition, AEDs also attributed to negative feelings such as being tired and sleepy as minor side effects which are narrated as follows

*"When I used it [AED] for the first time it made me a shock, feel tired and sleepy while I continue to use it I felt happy as it control my seizure and then I considered no life without using it".* 24-years-old female participant

**Positive feelings.** Despite these intense emotional distresses, some participants from urban residences who have strong social interaction (female, married, housewives and Male, grade, 12 students) reported that they were happy. The cause of happiness is not the disease itself but rather the social support and strong social interaction. In addition, their family and neighbours gave more attention to them which made them happy.

*"Thanks to God I am happier than others. I have a family. God gave me a lot of things to me. My husband made me to avoid doing work and my neighbors also help me by carrying out activities and taking care of my baby. I overcome the challenges with the help and advice of my husband"* a 24-years-old male participant

## Troublesome somatic symptoms

Symptoms began before seizure onset and continue after the occurrence of the seizure such as fatigue, feeling sleepy, stress, headache, and confusion were the most common experiences reported by study participants. The participants' narratives revealed that these uncomfortable physical experiences were attributed either due to the seizure or the effects of antiepileptic drugs.

For some young people with epilepsy, excessive fatigue was experienced only at the time of a seizure and might last for a short period, and, thus, they were then able to quickly return to normal classroom activities. For others, the persistent fatigue and need for sleep that accompanied their seizures could last hours or a whole day. More commonly, young people experienced fatigue as a continuous occurrence that at times was made worse by a seizure. Fatigue meant that young people needed more hours of sleep than usual.

*"I don't know what to say about my problem but I feel fatigued most of the time and then I fall down to the ground. I know everything that happened to me after I woke up"* 14- years-old, male

## Subtheme 2: Social isolation

The majority of participants in this study described social isolation as a dominant experience. Their sense of social isolation arose from (1) internal barriers and (2) external barriers.

**Internal barriers.** These include low self-esteem, lack of self-confidence, and feeling different.

Young people described personal conditions that affected their ability to participate in their social environment. They referred to having low self-esteem, lacking self-confidence, hesitating, and restraining themselves in social interactions as they experienced uncertainty

regarding their ability to be successful and to feel safe. Therefore, some young people identified feeling separate from, even different from, their peers and, therefore, restricting themselves from engaging fully in social activities. The following quote illustrates their emotional self-factors for their social interaction

*"When I move & talk with other people, I feel inferior to others. I think that most individuals do not consider me normal that is why I live alone and I hate living with family. Generally, I am intermediate between the life & the dead"* 22- years-old female, participant

**External barriers.**   Such as peers' exclusionary behaviours and Parent-imposed restrictions. Peers' exclusionary behaviours. Young people with epilepsy often suffered from exclusionary behaviors of their peers or their families. This took the form of being excluded from social activities, being teased and bullied by their peers, and being portrayed by their peers as different. The following quotes illustrate these hurtful experiences:

*"Currently I am silent and don't want to talk to any of my friends and families except dad and mom. I have been teased by most of my peers including my siblings. They consider me as different and they feared that my condition [epilepsy] will be transferred to them If they treat me when I have seizures."* *A 20-years-old male* participant

The majority of the young people with epilepsy in this study reported that they have difficulties engaging in a range of social activities such as Ethiopian cultural ceremonies as the society ignores and rejects them because of their being epileptic as illustrated in the following quote.

*"I have never been invited before in any ceremony after being ill because they [her family and neighbors] didn't remember me, they think I might have the illness at that moment in the ceremony. . .I consider people are not with me due to my illness [epilepsy] and also I didn't achieve my goal. . . Even if it is difficult to live alone, I am living alone to avoid being teased by neighbours of my family I am isolated from my family"* 22- years-old, male participant

## Subtheme 3: Parent-imposed restrictions

Some young people with epilepsy expressed excessive parental monitoring as the parents were excessively worried about them because of their seizures. Although young people viewed close parental monitoring as necessary because of safety concerns, they often felt frustrated by the restrictions that reduced their autonomy and opportunities to engage fully in social activities. This restriction was illustrated by the following narrative:

*". . .Even they [parents] do not leave me alone in transport taxi as well as holy water. I am 20 years old and I can follow my health condition but without my father, they do not allow to me even to come here [hospital] which made me angry and disappointed"* a 20-years-old male, participant

## Subtheme 4: Academic difficulties/challenges

Most of the young people whom I interviewed described some type of academic difficulty. Four of the participants had discontinued their learning from the primary level. Two of the

participants did not attend any form of school. They attributed the seizure or situations connected to it to these hurtful experiences. The following narratives clearly illustrate their academic difficulties either due to the seizure or situations connected to it such as using AEDs and memory impairment

> "*I forget everything including what I had learned and…... This condition [forgetfulness] was precipitated after I used two types of pills…in fact, currently, I discontinued my education…*"
> 19- years- old, a male participant

In addition, the tone of the narratives revealed that these young people with epilepsy had obstacles to their learning such as fatigue and difficulty in paying attention

> "*I am not good at my education, I think that is because I am especially lazy, and don't want to go to school every day even if I go to school, I cannot follow and catch up what a teacher said…... I just come home and fall asleep because I feel fatigued…*" 12- years- old male participant

In spite of these academic difficulties, none of the young people with epilepsy were enrolled in any special education classes or received extra help beyond the traditional classroom.

### Theme 2: Physical experience

**Subtheme 1: Recurrent physical trauma due to epilepsy.**   Half of the participants reported at least one form of physical injury experienced due to epilepsy. These include dental injury, minor head injury, tongue bite, and wound. Dental injury is the most frequently reported physical injury in this study. In addition, urine incontinence at the time of seizure in the public area was also reported by participants. This was disgusting and caused embarrassment to them. Such types of physical trauma led to intense emotional experiences including stress, and social embarrassment, and feel different as earlier expressed under the sub-theme; psycho-social.

> "*One day I was falling down to the ground and I lost my teeth (showed lost teeth). When I was in the holy water place, I was falling down and experienced bleeding from my tongue. I felt severe pain and even unable to eat food.*" 14-years-old female participant

### Theme 3: *Economical experiences*

Of eleven subjects, only one was in full-time employment at the time of the interview, four were still at school one female subject was a housewife, two were in daily labour and five were unemployed. These characteristics which were taken from socio-demographic data and the following narratives of young people with epilepsy in the direct and indirect cost of illness revealed that young people with epilepsy are an economically disadvantaged group as a result of their chronic illness.

### Subtheme 1: Direct cost of illness or out-of-pocket costs

A direct cost of epileptic care is attributable to specific services such as the cost of drugs, hospitalization, and transport costs. Although the majority of the participants reported that seizures can be controlled by inexpensive AEDs, some participants in this study described a profound treatment cost i.e expensive AEDs that are not available in the government hospital. Such

types of drugs are beyond their capacity to pay. For some participants from rural areas, transport cost is another economic burden. They are obliged to travel more than a hundred kilometres to see a doctor in a specialized and referral hospital which is beyond their capacity to pay.

"...*The price of the drug is also increasing now. I have no helper. I pay 2100 birr monthly for the drug [Lamotrigine]. I have no job I can't get this money... I travel more than a hundred kilometers to come here [Tibebe gion Specialized and Referral Hospital] I usually pay 75 birrs for single transport...*" 22- years-old female participant

## Subtheme 2: In direct cost of illness i.e. Productivity loss and unemployment/ job discrimination

The majority of the participants reported that achieving their economic independence was affected by either the seizures or hurtful experiences such as emotional negative feelings and restriction in activities as explained earlier under the psychosocial experiences. Most of the unemployed young people with epilepsy reported that having epilepsy made getting a job more difficult, either from personal experience or external constraints such as job discrimination. Many of the respondents who described difficulties while trying to obtain employment remarked that in most cases, their epilepsy had been the main reason why they had not succeeded. The following quotes revealed that young people had limitations in work activities.

"...*My friends are on a higher economic level; they all went to other countries[areas] and they returned with a lot of money but I could not do anything due to my illness[seizure]. even people are not volunteering to participate me in daily labor...*"19- years-old, male, participant

"*I cannot do anything; I fall down even in farmland activity....*"14-years-old, male, participant

"*... Leaders [employee] told me you are not competent enough for your job'...*" 23-year-old male respondent

## Theme 4: *Healthcare experience*

**Subtheme 1: Contexts related to Epileptic care.**   The usual prescribed anti-epileptic drugs were phenobarbital, phenytoin, carbamazepine, and lamotrigine. However, lamotrigine was not available in any of the referral hospitals, and there was no Carbamazepine in Tibebegion specialized and referral hospital at the time of data collection. Young people with epilepsy are obliged to seek and buy these drugs from a private pharmacy. There was a morning session in FHCSH and TGSH every healthcare professional started late their work. But at WCL3RH There was no morning session and every healthcare professional started and leave their work on time Health professionals working in the NCDs clinic were so busy and patients were required to wait more than four hours to get the service because of the daily follow-up patients were many in FHCSRH and TGSRH. But not in the WCL3RH psychiatric Clinic in which Epileptic patients get care services. The former two health facilities were overcrowded by patients. None of the health facilities did special activities other than prescribing medications for seizure control to those young people with epilepsy. These findings were obtained from health facilities' observational checklists. The above findings correspond with the in-depth interview results of this study.

*"Here [hospital] the first problem is the difficulty to know who comes first and who should get service first. Since there are a lot of patients, there is an incorrect patient card arrangement. They [health work forces] are willing to give first for their family or for whom they know. . .They do not give us psychological therapy. They [the hospital staff] think about only the monthly appointment not about us. Once upon a time, I came on my appointment and she [the nurse] said to me it is not your appointment it is for tomorrow. When I returned by the following day they said your appointment is passed. . ."* 15 -years–old, female, participant

The majority of the participants reported that the required information by young people with epilepsy on diagnosis and treatment options, side effects of medications, seizures and seizure control, injury prevention, psychological issues (especially stress), prognosis, and lifestyle were inadequate

*"They [health professionals] didn't inform me on treatment options, side effects of medications, stress, prognosis; they didn't explain anything about the disease [epilepsy]. Even they didn't investigate me with a machine [Electro encephalogram] . . ."* 22- years-old female participant

**Subtheme 2: Patient-provider relationship.**   The majority of the participants of this study reported health professionals to have bad approaches, disrespect, and insult their clients. One female participant reported physical violence i.e., slapping by a nurse working in the clinic. Such a type of health professional reaction leads to low patient satisfaction with the service provided in health facilities. The following quotes clearly illustrate health professionals' reactions to young people with epilepsy.

*". . .Four years back one of the nurses slapped me while I told her that I was the first comer. I was so upset at that time but I could not do anything. I remember still it was a hateful experience striking by a health professional. . .Currently; I do have seizures two-three times a week. When I talked about my illness to the health professionals, they said that 'we tried and tried about your illness [epilepsy]'. They [health professionals] don't treat us politely. They gave the medicine in high amounts. I feel tired and still, I am with the illness [seizure]. In general, it does not give me any satisfaction . . .. I also discontinued my learning."* 15 -years–old, female, participant

On the other hand, some of the participants reported that they got good services and that health professionals' approach was good to them. On the positive side of health professional reaction, the patients were satisfied with the health workforce as they had improvement from their seizures as illustrated by the following quote.

*"Here [in the hospital] they (the health professional) are trying their best to cure me. They gave me two types of drugs. And there is an improvement* in my illness. *I am happy with the health professionals and I am l satisfied with the services"* 19-years-old, male, participant

## Theme 5: Coping strategies toward epilepsy

The second main theme of coping strategies for epilepsy reflects the participants' efforts to balance their lives by developing strategies to solve problems related to their situation-related factors. This main theme had two subthemes: Seeking support and religious institutions and other tradition as a coping strategy

## Subtheme 1: *Seeking support*

Young people with epilepsy in this study seek both practical and emotional support. The majority of the participants used to find support from society and family members such as parents, peers, neighbours, and health professionals. This strategy involved getting help with physical problems from parents, reminders about taking medications, and maintaining good relationships with close friends to ensure support. Finding support from family members and society is an important and healthy strategy required to mitigate their hurtful experiences related to their illness and to ensure normal development. The following quote clearly illustrates support from family members such as her husband might result in her living a normal life with epilepsy

*"Thanks to my husband he always comes with me and helps me in every aspect of my illness without considering his job. . . After I gained my consciousness, I will have normal activity like any other person. Because of the disease, my husband and neighbor help me and they gave me special attention to me. That is why my social life became interesting"* 24- years- old female, participant

Getting practical support from injury at the time of seizure by peers is also another coping strategy to mitigate the physical experiences caused by the seizure itself or its hurtful psychosocial experiences such as suicidal ideation

*". . .Just by playing with my friends, I get satisfaction. . . I like them because they protect me from harming myself at the time of seizures and stress. . .When my friends let to relax and advise me as well as show me a film or go to a recreational area with me I feel happy and my condition gets improved. They also remind me to take my pills but after I separate from them, I feel bad and my condition will exacerbate."* 23- years-old, male, participant

On the other hand, as it was explained earlier under psychosocial experiences some young people with epilepsy require support but society might ignore giving support to them.

*". . .sometimes I try to control my situations by playing and talking with others around me. But people around my living place do not give attention to me. when this occurs, I leave that place and go to another place. . ."* 19-years- old, male, participant

## Subtheme 2: Religious institutions and other traditional healers as a coping strategy

Some participants of this study reported that they also tried the traditional treatment for their seizures and its' psychosocial effects. They went to a religious institution and they used holy water as a treatment option in addition they also get psychological as well as spiritual support in Sunday school to increase their social activities and their social contributions. Some other participants also reported that they had other traditional treatment options by going to traditional healers (sorcerers). The following quotes reveal such types of coping strategies

*"I used holy water in a religious place. It has some changes for the time being. I am also a member of Sunday school in which I got a lot of support like advice, having spiritual friends who protect me from injury when I have seizures"* 15- years-old, female, participant

*"Two months ago, for the interest of my mother I went to some cultural place (traditional healer) and he talked many, after he opened a book, gave me an unknown substance. . ."* 22-years-old, female participant

## Discussion

This study discussed the lived experience of young people with epilepsy in five interrelated themes. These include psycho-social experience, physical experience, economic experience, health care related experience and Coping strategies.

Young people who have negative feelings toward their chronic illness develop stress, embarrassment, fear of injury, and feeling different from their friends. This finding is in line with a study done in southern Ethiopia [24] that revealed more than one-third of people with epilepsy screened positive for psychological distress.

In addition, depression and anxiety with suicidal ideation are also serious negative feelings that reported from some study participants. This is also similar to a study done in central Ethiopia [25,26] in which poor social support, drug treatment for mental illness, depression, and uncontrolled seizures were significantly associated with such serious psychological feelings as suicidal ideation. The psycho-social findings of this study are also similar to the previous reports [27] in which social embarrassment and stigma, social isolation, and limitation in social interactions were some of the issues raised in it.

On the other hand, some participants reported that they had no psychological distress. This finding is similar to a study done in Northwest Ethiopia [28] and in Addis Ababa, Ethiopia [29]. This might be due to adolescents with epilepsy who had sufficient psycho-social support from families, friends and elders tended to experience less psychological distress. This indicate that psycho-social support from families, friends, teachers and health professionals is very important to increasing the negative feeling control of adolescent with epilepsy.

According to this study, social isolation is another profound hurtful psychosocial experience for young people with epilepsy. The majority of participants reported that the contextual factors for the social isolation were either internal factors such as low self-esteem, low self-confidence, and feeling different or external factors such as peers' exclusionary behaviours and parent-imposed restrictions. The last result is consistent with study in Singapore in which parent-imposed restriction was part of parents' attempts to protect them from harm [30].

In contrast, a study conducted at United States [31] revealed that the adolescent girls with epilepsy described themselves as normal teenagers. This difference might be due to cultural setting differences and the socio-economic differences between developing and developed nations. The previous report with adults had described the impact of culture on people's beliefs about social functioning, illness, and treatments [32]. These are very important factors in order to develop high self-esteem, self-confidence, and self-identity. Taking into consideration the different influences that society has on young people living with epilepsy, there is a need to extend research on cultural and economic influences on young people's well-being.

However, the present study's peers' exclusionary behaviours because of a difference that is labelled as undesirable by a group reflected that the young people experience a social stigma. This finding is similar to the previous report of the same study done among United states young people with epilepsy that were rejected because of a difference that is labelled as undesirable by a group [31].

Academic difficulties are the other form of psychosocial experience that faced by young people with epilepsy. This finding is similar with study at United states [31]. This might be due to the seizure or situations connected to either the seizure or the AEDs such as memory impairment, fatigue, and difficulty in paying attention.

Physical injury due to epilepsy is one of the major findings of this study. This finding is in line with a study done at Gondar [33] and England [34] where people with epilepsy in which PWE faced a number of physical injuries such as burn, fracture, dental loss, and haemorrhages

affecting the quality of patients' life to the extent of death. This might be due to young people with epilepsy are at greater risk of sustaining physical injury.

According to this study, young people with epilepsy experienced direct costs i.e. out-of-pocket costs, and indirect costs–productivity loss and unemployment. Seizures and their effects played a role in these experiences. They suffered job discrimination because of the disease. This is in harmony with the previous report [35]. This finding is in contrast with other study where unemployment rate due to epilepsy was lower for surgically treated patients than for non-surgically treated patients [36]. This difference might be due to surgical treatment has good outcome than medical treatment despite there is no surgical treatment center in our set up for epileptic patients.

In this study, the majority of participants had negative feedback regarding their previous epilepsy clinic health professional-patient relationship (e.g., insulting, disrespecting, slapping. . .). This finding is in contrast with a study done in Australia [37]. This difference might be due to the high workload in Ethiopian government hospitals and the low motivation of health professionals to give compassion and respectful care to their clients.

According to this study, the dissatisfaction of young people with the service provided as a result of poor patient-provider relationship in which inconsistent with a study done in developed countries [38] and Italy [39]. This disparity might be due to improper reaction of the health workforce towards young people with epilepsy, accessibility issues in which few specialized referral hospitals, poor integration of epileptic care to primary health care, and lack of health information contribute to poor satisfaction of young people with epilepsy towards the service provided.

In the present study, the majority of the participants used to find support from society and family members such as parents, peers, neighbours, health professionals, and siblings. Finding support from family members and society is an important and healthy strategy to counterbalance the effects of stigmatization on peer relationships and to ensure normal development, this also corresponds with a study done in United Kingdome that reported even though young people with epilepsy encountered stressful circumstances, they have positive adaptation by family processes including family connectedness, shared family beliefs, and communication processes that supported collaborative problem-solving [30].

According to this study, some of the participants reported that they also tried religious institutions and other traditions as coping strategies including using holly water as a treatment option and Sunday school participation to increase the social relations of young people with epilepsy. This is important to keep their spiritual health and be hopeful for the future. Keeping spirituality might be a protective factor against stress. This is similar to a study done in Ghana [40] but this finding might not be found in previous reports of western similar studies [30,31,41,42]. This might be because of cultural differences in western countries in which the function of religious institutions as a social connection bond and spiritual organization is weak.

In this study, some participants also reported that they used at traditional treatment options in addition to the AEDs by going to magicians or traditional healers. This is in harmony with the findings in Tanzania [43] and Sudan [44] in which young people with epilepsy used two or more treatment options for epilepsy, including faith healers and traditional healers, in addition to taking AEDs. Such types of findings were a reflection of their beliefs about the causes of epilepsy among the participants [40]. This might reduce their adherence to biomedical or modern medicine. Therefore, there is a need to increase education about epilepsy. However, these findings might not be found in Western similar studies [30,41]. This disparity might be due to cultural differences and different beliefs about the cause of epilepsy

### Limitation

The limitation of the study was the inability to generalize the findings of this qualitative interview study, but there are opportunities to transfer the findings to similar context and groups of people. Since this study is facility-based it might not explain the real experience of young people with epilepsy who did not have follow in health facilities. Another possible limitation of this study might be that the translated data of this study was not checked with an English language expert. Even though diversity should be a goal of the sampling process, it may never be fully achieved.

### Conclusions

In this qualitative interview study, the lived experiences of young people with epilepsy were addressed in terms of psychosocial, physical, economical, and healthcare-related experiences due to living with epilepsy. The identified psychosocial experience includes feeling about living with epilepsy (both hurtful negative and positive feelings), social isolation (caused by internal i.e., low self-esteem, lack of self-confidence, feeling different and external factors i.e., peers' exclusionary behaviour, parental imposed restriction), and academic difficulties/ discontinuing learning. The physical experiences also include physical injury due to epilepsy and troublesome somatic symptoms. The economic experiences cost of illness (direct and indirect) Finally, healthcare-related experiences were explored in terms of epileptic care context and health professional relationship with young people (disrespecting, insulting, slapping. . .) which results in dissatisfaction with the service provided were reported. In addition, coping strategies to wards epilepsy involving seeking support from family and society as well as religious institutions and other traditional sources of support were also identified.

Even though young people with epilepsy had suffered a number of hurtful experiences in psychosocial, physical, economic, and health care aspects due to living with epilepsy, they also reported coping strategies towards epilepsy that include support from various sources (family, neighbours, and religious institutions). Such narratives showed that young people with epilepsy's experiences were not always perceived to be negative. These types of findings have implications for social work interventions for young people living with epilepsy, in addition to prescribing AEDs and alleviating individual problems. Interventions can be directed at the utilization of the existing individual, family, and institution strengths to meet the challenges faced.

The purpose of this study was to explore lived experience of young epileptic patients. Hence, it is important for young epileptic patients in order to get comprehensive health care including psycho-social support and creating suitable working environment to overcome financial crisis. Future researchers should better conduct studies by using a mixed methodology for better understanding the quality of life of young people with epilepsy and including families/caregivers as potential participants to ensure young epileptic patients lived experience from families/caregivers' perspective.

### Supporting information

**S1 File.**
(DOCX)

### Acknowledgments

We would like to give our special thanks to the study participants for their contributions to this study. Our deepest gratitude goes to Felege-hiwot comprehensive specialized hospital

staff, Tibebe Gihon specialized hospital staff and West Command level 3 referral hospital staff for their assistance with patient recruitment.

## Author Contributions

**Conceptualization:** Kokeb Ayele, Habtamu Wondiye, Eyob Ketema Bogale.

**Data curation:** Kokeb Ayele, Habtamu Wondiye, Eyob Ketema Bogale.

**Formal analysis:** Kokeb Ayele.

**Methodology:** Kokeb Ayele, Habtamu Wondiye, Eyob Ketema Bogale.

**Software:** Kokeb Ayele, Habtamu Wondiye, Eyob Ketema Bogale.

**Supervision:** Habtamu Wondiye, Eyob Ketema Bogale.

**Validation:** Kokeb Ayele, Habtamu Wondiye, Eyob Ketema Bogale.

**Visualization:** Kokeb Ayele.

**Writing – original draft:** Kokeb Ayele, Habtamu Wondiye, Eyob Ketema Bogale.

**Writing – review & editing:** Kokeb Ayele, Habtamu Wondiye, Eyob Ketema Bogale.

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
