## [Decision Letter · Decision Letter 0]

8 Sep 2022

PONE-D-22-19800Lived Experience of Young People with Epilepsy in Bahir Dar City Government Hospitals, Ethiopia, 2021; Phenomenological studyPLOS ONE

Dear Dr. Bogale,

Thank you for submitting your manuscript to PLOS ONE. After careful consideration, we feel that it has merit but does not fully meet PLOS ONE’s publication criteria as it currently stands. Therefore, we invite you to submit a revised version of the manuscript that addresses the points raised during the review process.

Please note that we have only been able to secure a single reviewer to assess your manuscript. We are issuing a decision on your manuscript at this point to prevent further delays in the evaluation of your manuscript. Please be aware that the editor who handles your revised manuscript might find it necessary to invite additional reviewers to assess this work once the revised manuscript is submitted. However, we will aim to proceed on the basis of this single review if possible.

The reviewer has raised a number of concerns that need attention. They request additional information on methodological aspects of the study, revisions to the statistical analyses and they question the internal and external validity of the results reported.

Could you please revise the manuscript to carefully address the concerns raised?

We look forward to receiving your revised manuscript.

Kind regards,

Sebastian Shepherd

Staff Editor

PLOS ONE

Journal Requirements:

2.In the ethics statement in the Methods, you have specified that verbal consent was obtained. Please provide additional details regarding how this consent was documented and witnessed, and state whether this was approved by the IRB

Reviewers' comments:

Reviewer's Responses to Questions

**Comments to the Author**

1. Is the manuscript technically sound, and do the data support the conclusions?

Reviewer #1: Partly

2. Has the statistical analysis been performed appropriately and rigorously? 

Reviewer #1: N/A

3. Have the authors made all data underlying the findings in their manuscript fully available?

Reviewer #1: Yes

4. Is the manuscript presented in an intelligible fashion and written in standard English?

Reviewer #1: Yes

5. Review Comments to the Author

Reviewer #1: PONE-D-22-1800 Lived Experience of Young People with Epilepsy in Bahir Dar City Government Hospitals, Ethiopia, 2021; Phenomenological study

1. The study presents the results of original research.

Yes, this is original research.

2. Results reported have not been published elsewhere.

Yes

3. Experiments, statistics, and other analyses are performed to a high technical standard and are described in sufficient detail.

High technical standard and in sufficient detail, but there is a need of clarification about the recruitment procedure, which is purposive?, seems more as a quantitative strategic sampling performed by whom? Saturation? could only be reached after analysis. Please, give us some more information.

It is also confusing with this sentence regarding the strategic sampling; This was done to be able to see the different forms of the lived experience of self-care among type 1 diabetic patients from different perspectives.

The aim was to explore the lived experience of in YPE Bahir-Dar city

government referral Hospitals for a deep understanding of their condition and the

challenges they face in their day-to-day life and the impact of disease on their,

psychosocial, physical, and economic health. The aim and interview performance seem to fit a content analysis, descriptive phenomenology does also describe but there is more focus on the phenomenon and your analysis is now thematic analysis.

The title is Lived Experience of Young People with Epilepsy in Bahir Dar City Government Hospitals, Ethiopia, 2021; Phenomenological study

So, this study is about a selected group from just one hospital and their experiences from 2021? Is this study about their lived experiences or is it an evaluation of certain aspects due to this hospitals’ treatment program?

Phenomenology is presented as the method as such with reference 59. On the list there is only 49 references

Data collection refer to reference 60—not on the list. Interview guide developed with help from ref 38 (article about Temporal lobe surgery and memory) and ref 52 -not on the list.

Could you please, present the interview guide. How many questions/topics were there? During the interview the interviewer had taken notes immediately during the interview, regarding body language, verbal, and nonverbal cues, and the PI’s reflection of the interview?? The interview was conducted until conceptual saturation reached (to the point no further new information was obtained anymore). In addition, an observation checklist was used to collect care context data in health facilities. The interviews had a duration of 30-60 minutes. How was the interviewer able to perform all this actions and at the same time perform a phenomenological interview?

The analysis should be thematic analysis, there is no reference. The steps for this analysis are not presented. There is some more generic form of analysis, mixing unit meanings, categories, and themes. The result was then categories and predefined themes??

Trustworthiness is presented, according to reference 59, 61- not on the list. Much according to the book, not so much clarification about how you reached trustworthiness. Information about the interview guide and pilot interviewing should be placed at the data collection. Participants reviewing transcripts are not recommended, since the interview is spoken and then transcribed verbatim there are disparities to written text. Participants often are indignant about the language being so confusing. There is a need of clarification regarding all the components of trustworthiness in order to be able to follow the audit trail. How would audio recorder guarantee confirmability?

Correct information at the right place

The result- information about the participants can be placed in the methodology section—sample. Decide if the information should be in text or in table, now it is duplicated. In the table there are 10 participants not 11 regarding age.

The result is five 5 themes, Psycho-social experience, physical experience, economic experience, Health care related experience and Coping strategies towards epilepsy. All themes have two to four subthemes. The figure 1 is not correct there is themes and sub-themes—no categories. The experience is from young people with epilepsy- not all people with epilepsy. Presenting the theme, it seems as there are speculation and not just description of the data. Looking at the sub-themes there is not only description- there seem to be interpretation and ideas from the literature. There is description, interpretation adding written data, observations, and literature. There is a need to clarify and give references for what kind of analysis there has been. There is also a need to re-analyse the data, there should probably be two or three well balanced themes.

There are only quotations from 8 participants, but some have 2-3 quotations. How come?

This study is focusing on an important and interesting area of research, so if the analysis were analysed with a chosen methodology there would be interesting information.

The discussion is repeating the result and confirmed by references used, nothing new presented, but this could be due to poor analysis.

Limitations are well presented but are lacking methodology issues.

I have some ethical concerns-the participants are recognizable in the way they are presented in result section- table and quotations.

4. Conclusions are presented in an appropriate fashion and are supported by the data.

Conclusions are presented in appropriate fashion, probably supported by data, it is a repeated result in short-short version.

There is so much more in data, but now it is sorted into predefined themes?

5. The article is presented in an intelligible fashion and is written in standard English.

Yes, the article is presented in an intelligible fashion, but the structure could be sharpened, and it is written in standard English, mostly.

6. The research meets all applicable standards for the ethics of experimentation and research integrity.

Yes, this study meets all the applicable standards for research integrity except from presentation of 11 participants.

7. The article adheres to appropriate reporting guidelines and community standards for data availability.

Yes, the article is following the reporting guidelines.

This could be an interesting paper, presenting important knowledge. This paper needs to be re-worked, re-analysed and re-written presented with correct methodology then it would be suitable for publication.

Out of 49 references 20 are 10 years old or more, there are 23 references in the introduction and out of these 9 are 10 years old or more.

There are more actual articles about epilepsy, psychological aspect etc, should be added in the introduction. There is also a need to present the health care system regarding epilepsy in Ethiopia- this will lead us as readers forward to the aim and also give better understanding about the result

6. PLOS authors have the option to publish the peer review history of their article (what does this mean?). If published, this will include your full peer review and any attached files.

Reviewer #1: No

---

## [Author Response · Author response to Decision Letter 0]

31 Oct 2022

RESPONSE TO EDITOR AND REVIEWERS

RESPONSE TO EDITOR

RESPONSE: Thank you for coordinating the review process and fruitful comments. We have revised the manuscript and addressed Reviewer’s comments.

Journal Requirements: 

COMMENT: When submitting your revision, we need you to address these additional requirements.

RESPONSE: We have checked and attest that all formatting and style requirements have been met and revised based on the guideline 

COMMENT: 2. In the ethics statement in the Methods, you have specified that verbal consent was obtained. Please provide additional details regarding how this consent was documented and witnessed, and state whether this was approved by the IRB

RESPONSE: we have assured for you that ethical approval for this study was obtained from the Institutional Review Board of Bahir Dar University (Approval number: CMHS/IRB/ 26/008/2021).The Institutional Review Board of College of Medicine and Health Sciences, Bahir Dar University decided and approved that verbal informed consent obtained from each study participant could be enough to be ethically assured of the research process. This was because unless the name and the participants' medical registration number (MRN) were used during data collection, there is no ethical issue that will be raised. 

For young participants whose age was younger than 18 verbal informed consent was obtained from both participants and their legally authorized representatives before the study. Legally authorized representatives were given detailed information about the purpose of the study, data collection procedures, and possible risks and benefits of participating in the study through the consent process. Verbal informed consent was obtained from all legally authorized representatives whose children participated in the study. In this case, the families were presented as legally authorized representatives. A child was included in the study only if the families agreed with the child. Despite the families, consent, a child’s decision not to participate in the study was respected. 

Verbal informed consent was obtained from each study participant before the commencement of data collection. Participants also gave their verbal informed consent to share their data purely for scientific purposes without disclosing their true name. 

COMMENT: 3. In your Data Availability statement, you have not specified where the minimal data set underlying the results described in your manuscript can be found. PLOS defines a study's minimal data set as the underlying data used to reach the conclusions drawn in the manuscript and any additional data required to replicate the reported study findings in their entirety. All PLOS journals require that the minimal data set be made fully available. Important: If there are ethical or legal restrictions to sharing your data publicly, please explain these restrictions in detail. We will update your Data Availability statement to reflect the information you provide in your cover letter.

RESPONSE: Thank you for your fruitful comment, we upload it as supporting information

COMMENT: 4. Your ethics statement should only appear in the Methods section of your manuscript. If your ethics statement is written in any section besides the Methods, please move it to the Methods section and delete it from any other section. Please ensure that your ethics statement is included in your manuscript, as the ethics statement entered into the online submission form will not be published alongside your manuscript. 

RESPONSE: We have checked and ensured that ethical statement only appear in the methods section of our manuscript

Additional Editor Comments:

COMMENT: Please ensure the manuscript has undergo proof reading before resubmission 

RESPONSE: We have done it before resubmission.

RESPONSE TO REVIEWER 1:

Comments to the Author

5. Review Comments to the Author

RESPONSE: We thank the reviewer for kind words. We have revised our manuscript based on the comments as described below.

REVIEWER COMMENT: 3. Experiments, statistics, and other analyses are performed to a high technical standard and are described in sufficient detail. High technical standard and in sufficient detail, but there is a need of clarification about the recruitment procedure, which is purposive? seems more as a quantitative strategic sampling performed by whom? 

RESPONSE: Thank you for your fruitful comment, we have revised it as

A heterogeneous type of purposive sampling strategy was used to recruit the participants. Purposeful sampling is a technique widely used in qualitative research for the selection of information-rich cases, individuals that are experienced with a phenomenon of interest. In addition, it considers the ability to communicate experiences and opinions in an articulate, expressive, and reflective manner [15].

A heterogeneous type of purposive sampling strategy was used to recruit the participants. The heterogeneous characteristics of participants were maintained by considering age, marital status, educational status, health facilities, and residence, including the most vulnerable young people in the context of Ethiopian priorities and culture. This was done to be able to see the different forms of the lived experience of young epileptic patients from different perspectives.

We made a list of the characteristics (heterogeneous) that our participants should have. Then we Identified and sampled every person who meets the sample criteria. The doctors and nurses working in the follow up clinics and admission ward at Felege-Hiwot comprehensive specialized hospital (FHCSH), Tibebe Gihon specialized hospital (TGSH), and West Command level 3 referral hospital helped in identifying patients that fulfill the inclusion criteria of the study. Then the participants were communicated with before or after they get treatment on follow-up or admitted room after they recovered from their condition.

REVIEWER COMMENT: Saturation? could only be reached after analysis. Please, give us some more information.

RESPONSE: Thank you for your fruitful comment, we have revised it as saturation could not only be reached after analysis. It also maintained through data collection process.

The process of data collection continued until it reached the point of saturation. In turn, Data saturation was attained when all questions were answered, responses were completed and no further new information was obtained from various respondents. 

REVIEWER COMMENT: The title is Lived Experience of Young People with Epilepsy in Bahir Dar City Government Hospitals, Ethiopia, 2021; Phenomenological study So, this study is about a selected group from just one hospital and their experiences from 2021? Is this study about their lived experiences or is it an evaluation of certain aspects due to this hospitals’ treatment program?

RESPONSE: Thank you for your fruitful comment, this study is from 3 governmental hospital which listed in study setting. This study focuses on lived experiences of young epileptic patients and it is not evaluation of certain aspects due to this hospitals’ treatment program. But one of their experiences is compliance with the treatment at the health care setting. The participant shares their whole lived experience since they diagnosed with epilepsy and it is not their experiences from 2021.

REVIEWER COMMENT: Phenomenology is presented as the method as such with reference 59. On the list there is only 49 references

RESPONSE: Thank you, we have revised it as Reference 17. Guba EG, Lincoln YS. Competing paradigms in qualitative research. Handbook of qualitative research. 1994;2(163-194):105.

REVIEWER COMMENT: Data collection refer to reference 60—not on the list. 

RESPONSE: Thank you for your fruitful comment, we have revised it as Reference 18. Tolley EE, Ulin PR, Mack N, Robinson ET, Succop SM. Qualitative methods in public health: a field guide for applied research: John Wiley & Sons; 2016.

REVIEWER COMMENT: Interview guide developed with help from ref 38 (article about Temporal lobe surgery and memory) and ref 52 -not on the list.

RESPONSE: Thank you for your fruitful comment, we have revised it as Reference 19. Chen HJ, Chen YC, Yang HC, Chi CS. Lived experience of epilepsy from the perspective of children in Taiwan. Journal of clinical nursing. 2010;19(9‐10):1415-23.

20. Eklund PG, Sivberg B. Adolescents' lived experience of epilepsy. Journal of Neuroscience Nursing. 2003;35(1):40-50. 

REVIEWER COMMENT: Could you please, present the interview guide. How many questions/topics were there? 

RESPONSE: Thank you for your fruitful comment, there are 12 socio demographic related questions and 17 epileptic patients experience related questions.

DEMOGRAPHIC INFORMATION QUESTIONNAIRE (English version)

1.Age……………….

2. Sex……………….

3.Marital status:..........................

4.Educational status.....................

6. Religion ………………………..

8. Occupational status

9. How many years have you been epileptic? __________

10. Do you have history of admission in health institutions for epilepsy----

11. Where is your Residence rural/urban? -------

12. Number of AEDs you take….

Semi-structured in-depth interview guide (English version) 

Project Name:------------------------------ Interviewer:---------------------------------------------

Fake Name of Participant (ID):----------------------------- Date:------------------------------------

Start Time:-----------------End Time:------------------Location:-------------------------------------

Interview probes for young people with epilepsy 

1. What does to live with epilepsy mean to you?

2. How would you describe your experience of living with epilepsy? In terms of:

Probe: emotional, social, economic, physical and health care service-related situations. Can you give me an example of when that happened and how it happened? What did it feel like for you? How would you explain it or make sense of it 

3. What situations experienced connected with your life due to having epilepsy

4. When you think about having epilepsy, what emotions do you feel?

5. How do you feel when you have a seizure? What do you do at that time? How do you think your teachers and classmates look at your illness? How do you feel about the teachers’ and peers’ responses

How do you overcome those negative emotions related with your epilepsy condition probe: Emotional control, Emotional expressivity, Control over unpleasant thoughts, 

Stress management, Willpower and others

6. Please describe how your illness influences your social life. 

 Probe: Social contribution, social acceptance, social support, Personal relationship social intimacy social functioning, Self-esteem, Autonomy, Satisfaction with life roles. Would you tell me about your school life; how do you get on with your teachers, schoolmates and your courses? Please talk about how you get along with your parents and siblings . How do people treat you when they find out you have epilepsy?

7. What coping mechanisms do you use at time of challenges in social life? 

8. How do you describe your physical experience related with epilepsy? 

Probe: somatic complaints, Mobility, problems with performing activities of daily living

9. Could you tell me about any physical injury you have faced during seizure

Probe: Tongue bite, burn, other body injury and getting wet during seizure

10. How you overcome challenges you faced due to being an epileptic patient(related with physical experience)

11. How do you describe your economic status related with epilepsy condition

Probe: Satisfaction with living conditions (for example, financial situation) your income, debt, job loss, the cost you incur for treatment of epilepsy and opportunity cost 

12. What has been your experience with this health facility?

13. What did having epilepsy mean to you before you were diagnosed? Can you tell me about the medical support you received? What did it mean to be diagnosed with epilepsy?

14. How do you describe the treatment you received in health facility related your condition? 

Probe: utilization AED, psychotherapy, other non-medical support, during admission or OPD, challenges you faced in health facility and your satisfaction with service 

15. What do you think can be done for PWE? Would you ever consider changing your medication or your dosage and why is that?

16. How do you overcome challenges you faced in health facility?

17. Is there anything else you would like to share with me? Anything you think I should know about having epilepsy and traditional way of treatment for PWE

Thank participant for participating in the interview.

REVIEWER COMMENT: During the interview the interviewer had taken notes immediately during the interview, regarding body language, verbal, and nonverbal cues, and the PI’s reflection of the interview?? The interview was conducted until conceptual saturation reached (to the point no further new information was obtained anymore). In addition, an observation checklist was used to collect care context data in health facilities. The interviews had a duration of 30-60 minutes. How was the interviewer able to perform all this actions and at the same time perform a phenomenological interview? 

RESPONSE: Thank you for your fruitful comment, we would like to inform you that the interviewer uses audio recorder to record participant voice and he only take notes on nonverbal communication. Since it is in-depth interview the presence of others limit participant to tells their actual experience. So that the in-depth interview takes place without presence of others. In addition to this, the observation and in-depth interview takes place in the different time.

REVIEWER COMMENT: The analysis should be thematic analysis, there is no reference. The steps for this analysis are not presented. There is some more generic form of analysis, mixing unit meanings, categories, and themes. The result was then categories and predefined themes??

RESPONSE: Thank you for your fruitful comment, we have revised it 

Data analysis: After data collection, the principal investigator transcribed the audio-recorded data in the participant’s local language in written form and then back-translated to English by a third party, and then the data was coded, categorized, and analyzed based on the breadth, depth context, and nuance for ease of interpretation (thick description) (21). Data obtained from observation were written after completion of each fieldwork every day. The thematic analysis approach was used to analyse the data.

The principal investigators were read and reread the transcriptions several times and hear the audio-taped interview repeatedly to provide a sense of integrity and understand the meaning of the experiences from the participant's viewpoint. Each meaning unit was labeled with a code representing its content by open coding and then similar code organized into categories. Atlas. ti software version 7 was used to facilitate data analysis. Two independent coders were participated During coding. Categories were peer-reviewed and checked by the co-lead author and final subthemes and themes were created. Lastly, the report was written as subthemes and themes based on objectives for presenting the discoveries of the study. Quotes were used to highlight each category and show association with each theme. 

REVIEWER COMMENT: Trustworthiness is presented, according to reference 59, 61- not on the list. Much according to the book, not so much clarification about how you reached trustworthiness.

RESPONSE: Thank you for your fruitful comment, we have revised it reference as 17 and 21. We also provide much clarification about how we reached trustworthiness.

REVIEWER COMMENT: Information about the interview guide and pilot interviewing should be placed at the data collection.

RESPONSE: Thank you for your fruitful comment, we have revised it as

Literature was reviewed to develop an interview guide (19, 20). The data were collected using pretested interview guide. The interview guide contains 12 socio demographic question and 17 epileptic patients experience related questions. Interviews were conducted in the Amharic language. 

REVIEWER COMMENT: Participants reviewing transcripts are not recommended, since the interview is spoken and then transcribed verbatim there are disparities to written text.

RESPONSE: Thank you for your fruitful comment, we have removed it 

REVIEWER COMMENT: Participants often are indignant about the language being so confusing

RESPONSE: Thank you for your fruitful comment, it means the interview conducted in Amharic language which is the participants mother tongue language. 

REVIEWER COMMENT: There is a need of clarification regarding all the components of trustworthiness in order to be able to follow the audit trail. 

RESPONSE: Thank you for your fruitful comment, we have revised it as

Rigor and Trustworthiness of the Study: This study was trustworthy based on Lincoln and Guba’s criteria of credibility, dependability, conformability and transferability (17). 

Credibility: Credibility means ensuring the results are believable, consistent with reality and that the interpretations are true (21). Credibility was assured by keeping the consistency of procedures and the neutrality of the investigator about findings or decisions. Data collection sessions involved only those who are genuinely willing to take part and prepared to offer data freely (Ensure honesty in participants). Data collection tools were pretested before the actual study in a similar setting and population to ensure that the interview length and clarity of guide were appropriate for the actual participants. In addition to this, the interview guide was checked for consistency and correctness by an expert in a psychiatric clinic. Probing and multiple data sources (combination of in-depth interviews and observation) were employed to collect the data. Participants gave their verbal informed consent to share their data purely for scientific purposes without disclosing their true name. The questionnaires, audio data, transcriptions, and findings were given to the peer (who had qualitative research experience) to cross-check the processes, evaluate the findings and get feedback (peer debriefing). Moreover, at the onset of the study, bracketing the preconceptions of the investigator was employed to minimize the investigator’s bias and the risk of reactivity by the participants.

 Transferability: Transferability means showing that the findings have applicability in other contexts (21). To allow judgments about transferability by the reader appropriate probes had been used to obtain detailed information on responses. During each interview, the data collector observed participants' body movements; facial expressions, eye gaze, and tone of speech, and everything was recorded (persistent observations). Transferability was achieved by describing the study setting, sample, and data collection procedure clearly and using qualitative expert for peer debriefing. 

Dependability: Dependability is the assessment of the data collection and data analysis process. Dependability was attained through accurate documentation by minimizing spelling errors through frequent check, including all documents in the final report such as including the notes were written during the interview and ensure that the details of the procedures were described to allow the readers to see the basis upon which conclusions was made [21] each of which was considered at all times. 

Conformability: Conformability is a measure of how well the study’s discoveries are supported by the data collected and reflects the objectivity of the data. This means that the researcher’s bias should not alter the result (21). Confirmability of the study was ensured by the recording of every activity of the participants during the time of the interview. In addition, the audio-taped interviews were not destroyed which can enable others to track the process. Moreover, data analysis, interpretations and conclusions of findings were shared with experienced qualitative researchers for peer debriefing before synthesizing the final outputs. It was also achieved by using quotes which means linking the words of the participants with the discoveries (20).

REVIEWER COMMENT: How would audio recorder guarantee confirmability? Correct information at the right place

RESPONSE: Thank you for your fruitful comment, we have removed it and we write the confirmability as

Conformability: Conformability is a measure of how well the study’s discoveries are supported by the data collected and reflects the objectivity of the data. This means that the researcher’s bias should not alter the result (21). Confirmability of the study was ensured by the recording of every activity of the participants during the time of the interview. In addition, the audio-taped interviews were not destroyed which can enable others to track the process. Moreover, data analysis, interpretations and conclusions of findings were shared with experienced qualitative researchers for peer debriefing before synthesizing the final outputs. It was also achieved by using quotes which means linking the words of the participants with the discoveries (20).

REVIEWER COMMENT It is also confusing with this sentence regarding the strategic sampling; This was done to be able to see the different forms of the lived experience of self-care among type 1 diabetic patients from different perspectives.

RESPONSE: Thank you for your fruitful comment, we have removed such a confusing sentence and rephrase as 

 A heterogeneous type of purposive sampling technique was used to recruit the participants. The heterogeneous characteristics of participants were maintained by considering age, marital status, educational status, health facilities, and residence, including the most vulnerable young people in the context of Ethiopian priorities and culture. This was done to be able to see the different forms of the lived experience of young epileptic patients from different perspectives.

REVIEWER COMMENT: The aim was to explore the lived experience of in YPE Bahir-Dar city government referral Hospitals for a deep understanding of their condition and the

challenges they face in their day-to-day life and the impact of disease on their, psychosocial, physical, and economic health. The aim and interview performance seem to fit a content analysis, descriptive phenomenology does also describe but there is more focus on the phenomenon and your analysis is now thematic analysis.

RESPONSE: Thank you for your fruitful comment, our research study design is phenomenological study design and we have applied thematic analysis approach. There is no clear demarcation to use content analysis or thematic analysis. Based on our study design we have got thematic analysis best suited. Thematic analysis, as a popular form of qualitative data analysis helps in the identification of emerging patters from the set of events that one studies in content analysis. In content analysis we can also be interested to count verbs frequency to describe the interest of interviews on future aspects be motivated to explore relationships of words to see associations of ideas, or study the language of participants. They are very similar in the rules for coding. The reassuring thing is that both are applicably used within qualitative research as part of a process of meaning making and knowledge generation. 

REVIEWER COMMENT: The result- information about the participants can be placed in the methodology section—sample. Decide if the information should be in text or in table, now it is duplicated. 

RESPONSE: Thank you for your fruitful comment, we have revised it and we removed the text to reduce redundancy.

REVIEWER COMMENT: In the table there are 10 participants not 11 regarding age.

RESPONSE: Thank you for your fruitful comment, we have revised it. We have got editorial error for the age classification 20-24 years frequency and we revised it as 5

REVIEWER COMMENT: The result is five 5 themes, Psycho-social experience, physical experience, economic experience, Health care related experience and Coping strategies towards epilepsy. All themes have two to four subthemes. The figure 1 is not correct there is themes and sub-themes—no categories.

RESPONSE: Thank you for your fruitful comment, we have replaced categories with subthemes as follow: Figure 1: Themes and subthemes for lived experiences of Ethiopian young people with epilepsy, in Bahir-Dar city, Ethiopia, 2021.

REVIEWER COMMENT: - The experience is from young people with epilepsy- not all people with epilepsy. Presenting the theme, it seems as there are speculation and not just description of the data. Looking at the sub-themes there is not only description- there seem to be interpretation and ideas from the literature. There is description, interpretation adding written data, observations, and literature. There is a need to clarify and give references for what kind of analysis there has been. 

RESPONSE: Thank you for your fruitful comment, we have considered that there is a room to describe and interpret data in phenomenological study 

REVIEWER COMMENT: There is also a need to re-analyse the data, there should probably be two or three well balanced themes. 

RESPONSE: Thank you for your fruitful comment, we have seen the result part once again and we have found that there are no related themes to be merged. So that we keep all themes standing alone as it is. 

REVIEWER COMMENT: There are only quotations from 8 participants, but some have 2-3 quotations. How come?

Thank you for your fruitful comment, we have revised it again and tried to include from all 11 respondents.

REVIEWER COMMENT: - This study is focusing on an important and interesting area of research, so if the analysis were analyzed with a chosen methodology there would be interesting information. 

RESPONSE: Thank you for your fruitful comment, we have revised it again based on your comment

REVIEWER COMMENT: The discussion is repeating the result and confirmed by references used, nothing new presented, but this could be due to poor analysis.

RESPONSE: Thank you for your fruitful comment, we have revised it again based on your comment

REVIEWER COMMENT: Limitations are well presented but are lacking methodology issues. 

RESPONSE: Thank you for your fruitful comment, we have revised it 

Limitation: The limitation of the study was the inability to generalize the findings of this phenomenological study. No matter the phenomenon being investigated, the conclusions derived by the principal investigator apply only to that aspect of reality that was perceived by all participants. Despite cautions taken during the interviews, the introduction of social desirability bias in the descriptions of participants’ narration may be one of the possible limitations. Another limitation of this study was the introduction of recall bias at the time of the in-depth interview due to the chronic nature of the disease. Since this study is facility-based it might not explain the real experience of young people with epilepsy who did not have follow in health facilities. Another possible limitation of this study might be that the translated data of this study was not checked with an English language expert. 

REVIEWER COMMENT: I have some ethical concerns-the participants are recognizable in the way they are presented in result section- table and quotations.

RESPONSE: Thank you for your fruitful comment, we would like to notify for you that: - Participants also gave their verbal informed consent to share their data purely for scientific purposes without disclosing their true name. 

REVIEWER COMMENT: 4. Conclusions are presented in an appropriate fashion and are supported by the data. Conclusions are presented in appropriate fashion, probably supported by data, it is a repeated result in short-short version. There is so much more in data, but now it is sorted into predefined themes?

RESPONSE: Thank you for your fruitful comment, we have revised the conclusion by restating the purposes of the study with summarized results and conclusion that can early predict as follow: -

Conclusions: In this phenomenological study, the lived experiences of young people with epilepsy were addressed in terms of psychosocial, physical, economical, and healthcare-related experiences due to living with epilepsy. The identified psychosocial experience includes feeling about living with epilepsy (both hurtful negative and positive feelings), social isolation (caused by internal i.e., low self-esteem, lack of self-confidence, feeling different and external factors i.e., peers’ exclusionary behaviour, parental imposed restriction), and academic difficulties/ discontinuing learning. The physical experiences also include physical injury due to epilepsy and troublesome somatic symptoms. The economic experiences cost of illness (direct and indirect) Finally, healthcare-related experiences were explored in terms of epileptic care context and health professional relationship with young people (disrespecting, insulting, slapping…) which results in dissatisfaction with the service provided were reported. In addition, coping strategies to wards epilepsy involving seeking support from family and society as well as religious institutions and other traditional sources of support were also identified. 

Even though young people with epilepsy had suffered a number of hurtful experiences in psychosocial, physical, economic, and health care aspects due to living with epilepsy, they also reported coping strategies towards epilepsy that include support from various sources (family, neighbours, and religious institutions). Such narratives showed that young people with epilepsy’s experiences were not always perceived to be negative. These types of findings have implications for social work interventions for young people living with epilepsy, in addition to prescribing AEDs and alleviating individual problems. Interventions can be directed at the utilization of the existing individual, family, and institution strengths to meet the challenges faced.

The purpose of this study was to explore lived experience of young epileptic patients. Hence, it is important for young epileptic patients in order to get comprehensive health care including psycho-social support and creating suitable working environment to overcome financial crisis. Future researchers should better conduct studies by using a mixed methodology for better understanding the quality of life of young people with epilepsy and including families/caregivers as potential participants to ensure young epileptic patients lived experience from families/caregivers’ perspective.

REVIEWER COMMENT: 5. The article is presented in an intelligible fashion and is written in standard English. Yes, the article is presented in an intelligible fashion, but the structure could be sharpened, and it is written in standard English, mostly 

RESPONSE: Thank you for your kind words, we have revised our manuscript by correcting grammatical error.

REVIEWER COMMENT: 6. The research meets all applicable standards for the ethics of experimentation and research integrity. Yes, this study meets all the applicable standards for research integrity except from presentation of 11 participants. 

RESPONSE: RESPONSE: Thank you for your kind words, as we know qualitative sample size depend on level of saturation. So that our study reached saturation at 11 participants.

REVIEWER COMMENT: 7. The article adheres to appropriate reporting guidelines and community standards for data availability. Yes, the article is following the reporting guidelines.

RESPONSE: RESPONSE: Thank you for your kind words

REVIEWER COMMENT: This could be an interesting paper, presenting important knowledge. This paper needs to be re-worked, re-analyzed and re-written presented with correct methodology then it would be suitable for publication.

RESPONSE: Thank you for your kind words. We have revised it based on your suggestion.

REVIEWER COMMENT: Out of 49 references 20 are 10 years old or more, there are 23 references in the introduction and out of these 9 are 10 years old or more.

RESPONSE: Thank you for your kind words. We have revised it based on your suggestion by removing and replacing outdated references as much as possible as we can.

REVIEWER COMMENT: There are more actual articles about epilepsy, psychological aspect etc, should be added in the introduction. There is also a need to present the health care system regarding epilepsy in Ethiopia- this will lead us as readers forward to the aim and also give better understanding about the result

RESPONSE: Thank you for your fruitful comments. We have revised it based on your suggestion as:

People with epilepsy are also at increased risk of psychiatric comorbidity compared to the general population [14.]. A previous report stated that young people with epilepsy suffered poorer psychosocial outcomes compared to their peers. Young people with epilepsy had emotional suffering such as stress, angry, loneliness, embarrassment, annoyed, frustrated, sadness (15). 

In Ethiopia, a community study revealed that the majority of people with epilepsy were found to seek help from both religious and biomedical healing centers, with a 12-month treatment gap for biomedical care of 56.7% and a lifetime treatment gap of 26.9%. [16.].

---

## [Decision Letter · Decision Letter 1]

15 Nov 2022

PONE-D-22-19800R1Lived Experience of Young People with Epilepsy in Bahir Dar City Government Hospitals, Ethiopia, 2021 ; Phenomenological studyPLOS ONE

Dear Dr. Bogale,

Thank you for submitting your manuscript to PLOS ONE. After careful consideration, we feel that it has merit but does not fully meet PLOS ONE’s publication criteria as it currently stands. Therefore, we invite you to submit a revised version of the manuscript that addresses the points raised during the review process.

We look forward to receiving your revised manuscript.

Kind regards,

Nabeel Al-Yateem, PhD

Academic Editor

PLOS ONE

Journal Requirements:

Reviewers' comments:

Reviewer's Responses to Questions

**Comments to the Author**

1. If the authors have adequately addressed your comments raised in a previous round of review and you feel that this manuscript is now acceptable for publication, you may indicate that here to bypass the “Comments to the Author” section, enter your conflict of interest statement in the “Confidential to Editor” section, and submit your "Accept" recommendation.

Reviewer #1: All comments have been addressed

Reviewer #2: All comments have been addressed

2. Is the manuscript technically sound, and do the data support the conclusions?

Reviewer #1: Yes

Reviewer #2: Yes

3. Has the statistical analysis been performed appropriately and rigorously? 

Reviewer #1: N/A

Reviewer #2: Yes

4. Have the authors made all data underlying the findings in their manuscript fully available?

Reviewer #1: Yes

Reviewer #2: Yes

5. Is the manuscript presented in an intelligible fashion and written in standard English?

Reviewer #1: Yes

Reviewer #2: Yes

6. Review Comments to the Author

Reviewer #1: Thank you for all efforts in amending the manuscript. Most of the reviewer comments have been addressed. But there are still some issues to ameliorate. The result and the study performance are not a phenomenological study.

Some suggestions, in order to amend the manuscript.

The title; Lived Experience of Young People with Epilepsy in Bahir Dar City Government

Hospitals, Ethiopia, 2021; Phenomenological study-

change it to Lived Experience of Young People with Epilepsy in Bahir Dar City Government Hospitals, Ethiopia, A qualitative interview study

This study is about the lived experiences, so it is not necessary to point out 2021. (the period could be presented in data collection)

I am s ad to say, but this could never ever be a phenomenological study an interview guide with 14 demographic questions and then 17 epilepsy question- far too many to be a phenomenological study.

The interviews lasted between 30-60 minutes, with all these questions there is not an in- deep interview and the questions are directing the interviewee. But a qualitative interview study with a Thematic Analysis could be OK. There are no in-deep interviews, there is semi-structured interviews.

Revise it throughout the whole manuscript.

Delete data saturation below population and sample, you present it at the data collection section, which is a more suitable place,

Delete this sentence; Participants gave their verbal informed consent to share their data purely for scientific purposes without disclosing their true name, from the Rigor and trustworthiness—you have it un the Ethical section.

In the Discussion, you start with This study discussed the psychosocial distress of young people with epilepsy in three interrelated categories. These include feeling about living with epilepsy,

social isolation, and academic challenges. The result is presenting 5 themes???? So, what is this?

There is a need of correcting some text here in the discussion

Limitation, please don’t talk about bias in a qualitative study. What the interviewee is telling us, is what they have experienced, it is their reality. Yes, qualitative findings cannot be generalized but they can be transferable, Guba and Lincolns transferability, if we as readers can judge the audit trail and understand the process.

Methodological issues are about the researchers” bias,” maintaining neutral and being aware of own preconceived ideas.

Please re-write and correct this

The references are updated, but still there is no reference regarding phenomenology, so why not stay with the references—a qualitative interview study, analysed with thematic analysis approach.

I hope these comments will encourage you to make the amendments and thereby have a manuscript suitable for publication

I wish you the best

Reviewer #2: This is an interesting paper. I recommend to publish it with a minor revision.

Manuscript Number PONE-D-22-19800R1

Lived Experience of Young People with Epilepsy in Bahir Dar City Government Hospitals, Ethiopia, 2021; Phenomenological study

Comments and recommendation.

Thank you so much for giving me an opportunity to review this interesting paper. Below are my comments:

Design:

Population and sample:

1. Please explain how were study participants selected? It was purposive, convenience, etc.

2. I suggest to provide information regarding inclusion and exclusion criteria.

Data collection

3. In which interview data saturation was achieved?

4. Were transcripts returned to participants for comment or clarification?

Data analysis

5. Since it is a phenomenological study, please make clear the data analysis method. If thematic analysis, which thematic data analysis steps was adopted?

Results

6. In Theme 2 (Subtheme 2) - Troublesome somatic symptoms should be put in theme 1: Psycho-social experience

7. PLOS authors have the option to publish the peer review history of their article (what does this mean?). If published, this will include your full peer review and any attached files.

Reviewer #1: No

Reviewer #2: No

---

## [Author Response · Author response to Decision Letter 1]

16 Nov 2022

RESPONSE TO EDITOR AND REVIEWERS

RESPONSE TO EDITOR

RESPONSE: Thank you for coordinating the review process and fruitful comments. We have revised the manuscript and addressed Reviewer’s comments.

Journal Requirements: 

COMMENT: Please review your reference list to ensure that it is complete and correct. If you have cited papers that have been retracted, please include the rationale for doing so in the manuscript text, or remove these references and replace them with relevant current references. Any changes to the reference list should be mentioned in the rebuttal letter that accompanies your revised manuscript. If you need to cite a retracted article, indicate the article’s retracted status in the References list and also include a citation and full reference for the retraction notice.

RESPONSE: Thank you for your fruitful comment, we have reviewed our reference list and we can make sure for you that it our reference list is complete and correct

RESPONSE TO REVIEWER 1:

Comments to the Author

Reviewer #1: Thank you for all efforts in amending the manuscript. Most of the reviewer comments have been addressed. But there are still some issues to ameliorate. The result and the study performance are not a phenomenological study.

Some suggestions, in order to amend the manuscript.

RESPONSE: We thank the reviewer for kind words. We have revised our manuscript based on the comments as described below.

 REVIEWER COMMENT: The title; Lived Experience of Young People with Epilepsy in Bahir Dar City Government Hospitals, Ethiopia, 2021; Phenomenological study-

change it to Lived Experience of Young People with Epilepsy in Bahir Dar City Government Hospitals, Ethiopia, A qualitative interview study

RESPONSE: Thank you for your fruitful comment, we have revised our title by changing it by to Lived Experience of Young People with Epilepsy in Bahir Dar City Government Hospitals, Ethiopia, A qualitative interview study

REVIEWER COMMENT: This study is about the lived experiences, so it is not necessary to point out 2021. (The period could be presented in data collection)

RESPONSE: Thank you for your fruitful comment, we have revised it by removing 2021 from the title 

REVIEWER COMMENT: I am s ad to say, but this could never ever be a phenomenological study an interview guide with 14 demographic questions and then 17 epilepsy question- far too many to be a phenomenological study. The interviews lasted between 30-60 minutes, with all these questions there is not an in- deep interview and the questions are directing the interviewee. But a qualitative interview study with a Thematic Analysis could be OK. There are no in-deep interviews, there is semi-structured interviews. Revise it throughout the whole manuscript.

RESPONSE: Thank you for your fruitful comment, we have revised it as a qualitative interview study with a Thematic Analysis

REVIEWER COMMENT: Delete this sentence; Participants gave their verbal informed consent to share their data purely for scientific purposes without disclosing their true name, from the Rigor and trustworthiness—you have it un the Ethical section.

RESPONSE: Thank you for your fruitful comment, we have revised it by deleting that paragraph from the Rigor and trustworthiness.

REVIEWER COMMENT: In the Discussion, you start with This study discussed the psychosocial distress of young people with epilepsy in three interrelated categories. These include feeling about living with epilepsy,

social isolation, and academic challenges. The result is presenting 5 themes???? So, what is this? There is a need of correcting some text here in the discussion

RESPONSE: Thank you for your fruitful comment, we have revised it as

This study discussed the lived experience of young people with epilepsy in five interrelated themes. These include psycho-social experience, physical experience, economic experience, health care related experience and Coping strategies.

REVIEWER COMMENT: Limitation, please don’t talk about bias in a qualitative study. What the interviewee is telling us, is what they have experienced, it is their reality. Yes, qualitative findings cannot be generalized but they can be transferable, Guba and Lincolns transferability, if we as readers can judge the audit trail and understand the process.

Methodological issues are about the researchers” bias,” maintaining neutral and being aware of own preconceived ideas. Please re-write and correct this

RESPONSE: Thank you for your fruitful comment, we have revised it based on your comment. 

REVIEWER COMMENT: The references are updated, but still there is no reference regarding phenomenology, so why not stay with the references—a qualitative interview study, analyzed with thematic analysis approach.

RESPONSE: Thank you for your fruitful comment, we have revised it as a qualitative interview study and stayed with the references which support this paragraph

REVIEWER COMMENT: I hope these comments will encourage you to make the amendments and thereby have a manuscript suitable for publication

I wish you the best

RESPONSE: Thank you for your Kind Words

 RESPONSE TO REVIEWER 2:

Reviewer #2: This is an interesting paper. I recommend to publish it with a minor revision

RESPONSE: We thank the reviewer for kind words. We have revised our manuscript based on the comments as described below.

Design: Population and sample:

REVIEWER COMMENT: 1. Please explain how were study participants selected? It was purposive, convenience, etc.

RESPONSE: Thank you for your fruitful comments. We have explained it as follows:

A heterogeneous type of purposive sampling strategy was used to recruit the participants. Purposeful sampling is a technique widely used in qualitative research for the selection of information-rich cases, individuals that are experienced with a phenomenon of interest. In addition, it considers the ability to communicate experiences and opinions in an articulate, expressive, and reflective manner [21].

A heterogeneous type of purposive sampling technique was used to recruit the participants. The heterogeneous characteristics of participants were maintained by considering age, marital status, educational status, health facilities, and residence, including the most vulnerable young people in the context of Ethiopian priorities and culture. This was done to be able to see the different forms of the lived experience of young epileptic patients from different perspectives.

REVIEWER COMMENT: 2. I suggest to provide information regarding inclusion and exclusion criteria. 

RESPONSE: Thank you for your fruitful comments. We have provided inclusion and exclusion criteria as follows:

The participants were eligible for the study if they had been receiving care in the study clinic for at least 6 months duration since diagnosis. Seriously ill patients who are unable to communicate at the time of the data collection period were excluded from the study.

Data collection

REVIEWER COMMENT: 3. In which interview data saturation was achieved?

RESPONSE: Thank you for your fruitful comments. We have revised our manuscript by inserting the data about in which interview data saturation was achieved under subheading data collection as:

The data saturation was achieved in nineth interview and the additional two interviews were conducted for confirmation of data saturation.

REVIEWER COMMENT: 4. Were transcripts returned to participants for comment or clarification?

RESPONSE: Thank you for your fruitful comments. We have been invited some of the participants for transcription verification and we have revised it based on your suggestion as follow in credibility part of rigor and trustworthiness section:

We also invited 6 participants to review the ideas for transcription verification, which they think the investigator is going to present a true picture from their perspective.

Data analysis

REVIEWER COMMENT: 5. Since it is a phenomenological study, please make clear the data analysis method. If thematic analysis, which thematic data analysis steps was adopted?

RESPONSE: Thank you for your fruitful comments. Our data analysis method was thematic analysis. We have revised it based on your suggestion as:

The thematic analysis approach was used to analyse the data. The six thematic data analysis steps were followed to analyze the data such as Step 1: Become familiar with the data, Step 2: Generate initial codes, Step 3: Search for themes, Step 4: Review themes, Step 5: Define themes and Step 6: Write-up [23].

Results

REVIEWER COMMENT: 6. In Theme 2 (Subtheme 2) - Troublesome somatic symptoms should be put in theme 1: Psycho-social experience

RESPONSE: Thank you for your fruitful comments. We have revised it based on your suggestion by moving Troublesome somatic symptoms from Theme 2 (Subtheme 2) to theme 1: Psycho-social experience

---

## [Decision Letter · Decision Letter 2]

21 Nov 2022

PONE-D-22-19800R2Lived Experience of Young People with Epilepsy in Bahir Dar City Government Hospitals, Ethiopia, 2021;  Aqualitative interview  studyPLOS ONE

Dear Dr. Bogale,

Thank you for submitting your manuscript to PLOS ONE. After careful consideration, we feel that it has merit but does not fully meet PLOS ONE’s publication criteria as it currently stands. Therefore, we invite you to submit a revised version of the manuscript that addresses the points raised during the review process.

We look forward to receiving your revised manuscript.

Kind regards,

Nabeel Al-Yateem, PhD

Academic Editor

PLOS ONE

Journal Requirements:

Reviewers' comments:

Reviewer's Responses to Questions

**Comments to the Author**

1. If the authors have adequately addressed your comments raised in a previous round of review and you feel that this manuscript is now acceptable for publication, you may indicate that here to bypass the “Comments to the Author” section, enter your conflict of interest statement in the “Confidential to Editor” section, and submit your "Accept" recommendation.

Reviewer #1: All comments have been addressed

Reviewer #2: All comments have been addressed

2. Is the manuscript technically sound, and do the data support the conclusions?

Reviewer #1: Yes

Reviewer #2: Yes

3. Has the statistical analysis been performed appropriately and rigorously? 

Reviewer #1: Yes

Reviewer #2: N/A

4. Have the authors made all data underlying the findings in their manuscript fully available?

Reviewer #1: Yes

Reviewer #2: Yes

5. Is the manuscript presented in an intelligible fashion and written in standard English?

Reviewer #1: Yes

Reviewer #2: Yes

6. Review Comments to the Author

Reviewer #1: Thank you for your efforts. Most comments have been adressed. Just some small corrections:

In the abstracts: change, The data were collected through in-depth interview--to semi structured interviews

In limitations: add, The limitation of the study was the inability to generalize the findings of this qualitative interview study, but there is opportunities to transfer the findings to similar context and groups od people.

Reviewer #2: Thank you so much for your responses to my comments and suggestions. Good luck with your publication.

7. PLOS authors have the option to publish the peer review history of their article (what does this mean?). If published, this will include your full peer review and any attached files.

Reviewer #1: No

Reviewer #2: No

---

## [Author Response · Author response to Decision Letter 2]

21 Nov 2022

RESPONSE TO EDITOR AND REVIEWERS

RESPONSE TO EDITOR

RESPONSE: Thank you for coordinating the review process and fruitful comments. We have revised the manuscript and addressed Reviewer’s comments.

Journal Requirements: 

COMMENT: Please review your reference list to ensure that it is complete and correct. If you have cited papers that have been retracted, please include the rationale for doing so in the manuscript text, or remove these references and replace them with relevant current references. Any changes to the reference list should be mentioned in the rebuttal letter that accompanies your revised manuscript. If you need to cite a retracted article, indicate the article’s retracted status in the References list and also include a citation and full reference for the retraction notice.

RESPONSE: Thank you for your fruitful comment, we have reviewed our reference list and we can make sure for you that our reference list is complete and correct

RESPONSE TO REVIEWER 1:

REVIEWER COMMENT: Reviewer #1: Thank you for your efforts. Most comments have been addressed. Just some small corrections:

In the abstracts: change, the data were collected through in-depth interview--to semi structured interviews

RESPONSE: We thank the reviewer for kind words. We have revised it by replacing in-depth interview—by-semi structured interviews

REVIEWER COMMENT: In limitations: add, the limitation of the study was the inability to generalize the findings of this qualitative interview study, but there is opportunities to transfer the findings to similar context and groups of people.

RESPONSE: We thank the reviewer for kind words. We have revised our limitation section by adding your suggestion as:

The limitation of the study was the inability to generalize the findings of this qualitative interview study, but there are opportunities to transfer the findings to similar context and groups of people 

RESPONSE TO REVIEWER 2:

REVIEWER COMMENT: Reviewer #2: Thank you so much for your responses to my comments and suggestions. Good luck with your publication.

RESPONSE: We thank the reviewer for kind words and for your best wish.

---

## [Editor Report · Decision Letter 3]

24 Nov 2022

Lived Experience of Young People with Epilepsy in Bahir Dar City Government Hospitals, Ethiopia, 2021;  A qualitative interview  study

PONE-D-22-19800R3

Dear Dr. Bogale,

We’re pleased to inform you that your manuscript has been judged scientifically suitable for publication and will be formally accepted for publication once it meets all outstanding technical requirements.

Kind regards,

Nabeel Al-Yateem, PhD

Academic Editor

PLOS ONE
---

## [Editor Report · Acceptance letter]

2 Dec 2022

PONE-D-22-19800R3 

Lived Experience of Young People with Epilepsy in Bahir Dar City Government Hospitals, Ethiopia, A qualitative interview study 

Dear Dr. Bogale:

I'm pleased to inform you that your manuscript has been deemed suitable for publication in PLOS ONE. Congratulations! Your manuscript is now with our production department. 

Kind regards, 

on behalf of

Dr. Nabeel Al-Yateem 

Academic Editor

PLOS ONE